# Mucosal Immunity: Lessons from the Lower Respiratory and Small Intestinal Epithelia

**DOI:** 10.3390/biomedicines13051052

**Published:** 2025-04-26

**Authors:** Kayle B. Dickson, Andrew W. Stadnyk, Juan Zhou, Christian Lehmann

**Affiliations:** 1Department of Microbiology and Immunology, Dalhousie University, Halifax, NS B3H 4R2, Canada; kayle.dickson@dal.ca (K.B.D.); andrew.stadnyk@dal.ca (A.W.S.); 2Department of Pediatrics, Dalhousie University, Halifax, NS B3H 4R2, Canada; 3Department of Anesthesiology, Pain Management and Perioperative Medicine, Dalhousie University, Halifax, NS B3H 4R2, Canada; juan.zhou@dal.ca; 4Department of Pharmacology, Dalhousie University, Halifax, NS B3H 4R2, Canada; 5Department of Physiology and Biophysics, Dalhousie University, Halifax, NS B3H 4R2, Canada

**Keywords:** small intestinal epithelium, respiratory epithelium, mucosal immunity

## Abstract

Mucosal epithelia represent a diverse group of tissues that function as a barrier against the external environment and exert a wide variety of tissue-specific secondary functions. This review focuses on the lower respiratory tract and small intestinal epithelia, which serve as two distinct sites within the body with respect to their physiological functions. This review provides an overview of their physiology, including both physiological and mechanical defense systems, and their immune responses, which allow both tissues to tolerate commensal organisms while mounting a response against potential pathogens. By highlighting the commonalities and differences across the two tissue types, opportunities to learn from these tissues emerge, which can inform the development of novel therapeutic strategies that harness the unique properties of mucosal epithelia.

## 1. Introduction

Epithelial cells represent a diverse group of cells organized into tissues that line all surfaces of the body exposed to the external environment. Mucosal epithelia, including the respiratory tract, gastrointestinal (GI) tract, and urinary/reproductive tracts, line cavities of the body where they are continuous with the skin at the cavity opening, and cover the surfaces of some internal organs. The primary role of the mucosal epithelium is to serve as a barrier against the external environment, but individual epithelia have tissue-specific secondary functions. The immune responses at these epithelia, collectively referred to as mucosal immunity, are of particular interest as these sites are vulnerable to pathogens [1]. As such, mucosal epithelia possess elaborate mechanisms to prevent pathogen invasion and dissemination to other sites within the body. This review focuses on the lower respiratory tract and small intestinal epithelia, which represent two physiologically and functionally distinct sites within the body that face constant exposure to the external environment. This review provides an overview of their physiology, defense systems, and immune responses, highlighting their commonalities and differences. This comparison highlights opportunities to learn from these tissues, informing the development of novel therapeutic strategies that harness the unique properties of mucosal epithelia.

## 2. General Structure and Function

In general, mucosal epithelial cells serve as a barrier between self and non-self, preventing microorganisms, toxins, and other unwanted matter from entering the body. For example, both the lower respiratory and small intestinal epithelia are exposed to the external environment via respiration and digestive processes, respectively, at the apical side of the epithelium. The specific structure of the epithelia and their additional functions differ based on location, with phenotypic adaptations that support specific functions. The lower respiratory tract encompasses the lower portion of the larynx, trachea, and lungs, and can be further subdivided into the large and small airways, with the large airway being distinguished by the presence of cartilaginous support. The small intestine begins where the stomach is emptied into the duodenum and includes the jejunum and the ileum. These epithelia are distinct sites, and as such, they have unique adaptations to address tissue-specific exposures to microbes and other insults.

Epithelial tissues are broadly classified by organization and cell shape (e.g., squamous, cuboidal, or columnar). Simple tissues consist of a single layer of epithelial cells, while stratified epithelia (e.g., skin) have multiple layers of stacked cells, which are specifically adapted to withstand shear forces and tolerate the repeated sloughing of cells. Pseudostratified epithelia consist of a single layer of irregularly shaped cells, which appear stratified though all cells reach the basement membrane. Transitional epithelium is a sub-type of stratified epithelium found only in the urinary system, which has the unique ability to expand to accommodate distension of the bladder. Epithelial tissues may also form glandular structures where the epithelial cells project into the underlying tissues. For example, exocrine glands retain continuity with the epithelial tissues and can be unicellular or multicellular structures.

The organizational features of epithelia vary depending on location and are optimized to facilitate the function of that specific tissue. Additionally, epithelial cells can undergo significant changes throughout tissue morphogenesis, including changes in cell shape, arrangement, and division, to form the required tissue structure [2]. In the respiratory tract, the epithelium of the large airway consists primarily of pseudostratified columnar epithelium, which transitions to simple cuboidal epithelium in the small airway. In contrast, the small intestinal epithelium features simple columnar epithelial cells, which are organized into villi and crypts. The landscape of each epithelium is depicted in Figure 1.

Structural polarity, with clear structural and functional divisions between the apical and basolateral surfaces of the cell, is a critical aspect of epithelial cell organization. Tight junctions between adjacent cells have a “fence function” and represent the boundary between apical and basolateral surfaces [3]. Polarity is established by a complex integrin-mediated signaling pathway involving both inside-out signals to assemble the basement membrane assembly and outside-in signaling to establish microtubule orientation and vesicle trafficking pathways [4]. The establishment of polarity in unpolarized precursor cells relies heavily on mutually antagonistic interactions and positive feedback loops created by various polarity regulators, such as the Par, Crumbs, and Scribbles protein complexes [5]. The establishment and maintenance of polarity in epithelia are reviewed in detail by Buckley and Johnston [6].

As one key function of mucosal epithelium is to maintain adequate hydration levels in the body cavity while ensuring protection from the external environment, permeability is tightly regulated across all mucosal epithelia. Selectively permeable tight junctions mediate cell-to-cell adherence and act as a dynamic barrier at the apical membrane of epithelial cells, responding to signals from the immune system and other environmental stimuli [7]. The major classes of transmembrane proteins found in tight junctions are occludins, claudins, and junction adhesion molecules. Cadherin-based structures, such as adherens junctions and highly structured desmosomes, also facilitate cell-to-cell adherence by connecting to the actin cytoskeleton [8]. The diffusion of ions, electrical currents, and other small molecules are facilitated by connexins, which form gap junctions between cells. These structural features are relatively conserved across various epithelial tissues, with some adaptations to meet the functional needs of individual tissues [9]. As an example, epithelial permeability can be altered in response to perceived threats. In the respiratory epithelium, activation of bronchial cell Toll-like receptor 2 (TLR2) results in a concentration-dependent increase in claudin-1 expression and enhanced tight junction integrity [10]. Similar mechanisms exist in the intestinal epithelium to preserve barrier integrity [11].

## 3. Cell Populations

A variety of cell types are found within the epithelium of the respiratory tract and the small intestine. Within the small intestine, enterocytes make up 90% of the cell population [12]. These cells feature small, non-motile microscopic protrusions, termed microvilli, on the apical membrane that increase the surface area of enterocytes to facilitate nutrient absorption. Within the large airway portion of the lower respiratory tract, most cells are terminally differentiated, and non-secretory. These cells feature larger, motile microscopic protrusions, termed cilia, which play a key role in the mucociliary clearance of entrapped pathogens. As the respiratory epithelium transitions to a simple cuboidal structure within the small airway, the proportion of ciliated cells decreases [13]. These structures, along with additional specialized cell types in both epithelia, are illustrated in Figure 1.

Both epithelia possess additional specialized cell types, which often have comparable functions. Bioinformatic approaches, including single-cell sequencing, have been critical for the identification and description of rare cell types and novel phenotypes, resulting in an increased appreciation of the dynamic landscape of these epithelia. As summarized in Table 1, the cell populations in both epithelia can be divided into secretory and non-secretory cell types to facilitate cross-epithelial comparisons.

Importantly, the proportions of different specialized cell populations may differ substantially along the respiratory/intestinal tract, which correlates to the functional changes that occur along each epithelium. For example, goblet cells increase caudally along the GI tract, resulting in enhanced mucus secretion along the tract [14]. It is important to note that these proportions are not always identical in murine models, posing a challenge when studying epithelial populations in murine models [15]. Basal cell proportions decrease progressively throughout the small airway in humans but are completely absent beyond the trachea of the mouse, limiting the utility of the lower airway of mice for the study of cell populations in the human lower airway [16,17]. Advances in research methodology, including the use of organoids generated from samples of human primary epithelial cells, are poised to address this ongoing challenge [18].

### 3.1. Non-Secretory Cells

Stem cells play an important regenerative role in tissues that experience rapid cell turnover. The intestine has the most rapid rate of renewal of all epithelial tissues, experiencing complete turnover every 4–5 days [19]. Stem cells are located at the base of crypts within the small intestinal epithelium, where daughter cells of stem cells differentiate into immature enterocytes, which then mature as they migrate from the crypts to the tips of the villi [20]. In the case of injury, pro-inflammatory signals trigger rapid proliferation and epithelial cell migration within the crypt niche to preserve epithelial integrity [12]. In contrast to the small intestinal epithelium, ciliated epithelial cells in the trachea have a relatively slow rate of renewal, with a half-life of over six months [21]. Relatively undifferentiated basal cells within the respiratory epithelium possess stem cell-like properties and act as the primary progenitor cell [22].

Single-cell sequencing has increased our understanding of additional differentiation dynamics within the respiratory epithelium. This includes the identification of deuterosomal cells, which are a precursor to multi-ciliated cells, and express markers of both ciliated and goblet cells [23]. Centriole amplification occurs as a consequence of transcriptional reprogramming and expression of the atypical cyclin O, inducing a cell cycle variant that favors differentiation over mitosis [24]. Club cells can also act as progenitors for both ciliated and goblet cells [25,26]. This functional overlap can overcome the slow rate of proliferation within the respiratory epithelium to maintain the epithelial function in the case of damage. Similarly, keratin-13-positive (KRT13+) cells, referred to as Hillock cells when first discovered, are a transitional cell type that occur in high-turnover squamous epithelial sites within the respiratory epithelium. These cells express genes associated with squamous cell differentiation, cell adhesion, and immunomodulation, but their overall function remains poorly described [27]. A recent review by Lin et al. describes these cells as highly resistant to injury and able to regenerate a variety of cell types within the respiratory epithelium, contributing to the repair of denuded epithelial surfaces [28]. The immune functions associated with these cells are currently under investigation, but evidence suggests that they have a pro-inflammatory phenotype and may play a role in innate immune activation [29].

Both epithelia have chemosensory cells known as tuft/brush cells that can be distinguished morphologically by the “tuft” of microvilli on the apical surface of the cell. Despite their relatively low abundance in both the small intestine and lower respiratory epithelia, these cells can have functions beyond chemosensation, and represent an emerging area of immunology research [30,31]. Within both epithelia, two subsets of tuft cells have been identified. In the mouse small intestine, tuft-1 cells express genes related to neuronal development, while tuft-2 cells express immune-related genes [32]. Humans express four different types of tuft cells, which exhibit some overlap with their murine counterparts, while also demonstrating stem cell-like characteristics [31]. Similar classifications exist within the lower respiratory epithelium, with tuft-1 cells expressing genes related to taste and tuft-2 cells expressing genes related to the immune function (e.g., leukotriene biosynthesis) [27]. Current evidence suggests that these cells are highly heterogeneous with a diverse array of functions that are likely tissue and disease-specific [33]. Both respiratory and intestinal tuft cells have been described as sentinel cells, being able to respond to cues from micro-organisms. Specifically, tuft cells are able to mediate Type II immunity, in response to the helminth infection within the intestine, and to allergens in the lower respiratory tract [34,35].

Microfold (M) cells are a unique population found in both the lower respiratory and small intestinal epithelium that act as a bridge between innate and adaptive immunity. M cells are responsible for antigen delivery from the apical side of the epithelia to antigen-presenting cells, including immature dendritic cells (DCs), ultimately activating naïve T cells upon maturation. This process is critical for the induction of both tolerance to the local microbiome and immune responses to pathogens. M cells are present in low proportions in lymphoid-associated tissues within both epithelia, such as the epithelium overlying Peyer’s patches of the intestine [32]. While M cells can be found under homeostatic conditions in the large airway, in the small airway, lymphoid-associated tissues and M cells primarily develop during disease, and the differentiation of M cells relies on inflammatory signals, including RANKL and NF-κB [36,37]. M cells can also act as a portal of entry for some pathogens, including *Mycobacterium tuberculosis*, in the airway [38]. A similar phenomenon has also been described in the intestine, with various enteric pathogens using M cell transcytosis as a pathway through the epithelial barrier [39].

Of note, both the respiratory tract and small intestine have robust populations of intraepithelial lymphocytes, which are the first adaptive immune cells to encounter invading microbes within either epithelium. Though not entirely understood, these cells have a variety of roles in immune surveillance and host defense [40].

### 3.2. Secretory Cells

Goblet cells are the principal mucus-secreting cell within the two epithelia. Within the lower respiratory epithelium, goblet cells have two distinct phenotypes: goblet-1 cells primarily produce mucus, while goblet-2 cells secrete ZG16, a protein involved in the aggregation of Gram-positive bacteria [41]. Of note, these cells are abundant within the large airway, and decrease in density within the small airway [42]. Goblet cells are also found within the small intestinal epithelium where they serve a similar role in the production of mucus. The role of goblet cells and mucus production in both the respiratory and intestinal epithelia will be described in detail later in this article.

The lower respiratory epithelium contains additional non-mucus secreting cells, including club cells and pulmonary ionocytes. Club cells are found in higher proportions in the terminal bronchioles, where they secrete the anti-inflammatory uteroglobin protein (also known as club cell 10 protein) and other substances that act as pulmonary surfactants, effectively replacing mucus-secreting goblet cells in this tissue [43]. Some evidence also suggests a role for club cells in regulating oxidative stress and protease/anti-protease balance [44]. Single-cell techniques have allowed for the identification of pulmonary ionocytes, which play a role in ion transport, along with airway fluid pH and balance [23,45]. These cells are currently a hot topic in cystic fibrosis research, as they express considerably higher levels of the cystic fibrosis transmembrane regulator (CFTR) than other respiratory cells [46].

Paneth cells, highly specialized secretory cells, are located within the crypts of the small intestine. They play a key role in GI homeostasis, secreting both anti-microbial peptides (AMPs) and immunomodulatory proteins that regulate the composition of the intestinal microbiome [47]. Paneth cells are enriched within the ileum of the small intestine, where there is the greatest microbial burden [48]. Recent evidence suggests that various subtypes of Paneth cells may exist in the mouse small intestine, exhibiting distinct phenotypes and functions [49]. In their recent review, Quintero and Samuelson suggest that Paneth cells function largely as niche cells, which contribute to stem cell function, but are ultimately dispensable within the gut [50].

Endocrine cells are involved in the production and secretion of hormones and are found in both lower respiratory and small intestinal epithelia. The functions of these hormones vary substantially based on the tissue but many have immune activity. Enteroendocrine cells are traditionally divided into eight subtypes based on the hormones secreted, though a significant phenotypic overlap occurs [32]. Some of these subtypes localize preferentially along specific sections of the intestine. As reviewed by Atanga et al., the peptides and hormones secreted by enteroendocrine cells interact with both immune cells and the enteric nervous system, and can be linked to a variety of disease states within the intestine [51]. Within the respiratory epithelium, neuroendocrine cells play a critical role in facilitating interactions between the epithelium and nervous system. They secrete various molecules, such as serotonin, calcitonin gene-related peptide, and bombesin-like peptide, which interact with both epithelial cells and immune cells to alter proliferation, angiogenesis, immune cell chemotaxis, and cytokine expression [41].

## 4. Mechanical and Physical Defense Systems

As both the respiratory and small intestinal epithelia are constantly exposed to external stimuli, they have well-developed innate defense systems to prevent or control colonization by microbes. Both epithelia have mucus layers that serve as a physical barrier against microbes. In the respiratory epithelium, mechanical mucociliary clearance further supports the clearing of particulate matter that becomes trapped in the mucus.

Mucociliary clearance is the result of a coordinated effort between ciliated epithelial cells and mucus-producing goblet cells. Cilia reaching into the airway are covered with a thin layer of low-viscosity periciliary fluid, surfactant, and mucus to trap microbes [13]. Particle size determines the site of deposition, with particles exceeding 8 μm depositing in the large airways for mucociliary clearance, while smaller molecules (<8 μm) deposit in the small airway and must be managed with alternative defense mechanisms [52]. To facilitate removal, mucin production can be upregulated by exposure to pathogen-associated molecular patterns (PAMPs), pro-inflammatory cytokines, beta-neutrophil elastase, growth factors, and cigarette smoke [22]. Mucin MUC5AC responds rapidly to environmental insults, while MUC5B is involved in chronic infection/inflammation [53]. This represents an adaptive response to immune threats; overexpression of MUC5AC in a murine model reduced influenza infection levels and decreased neutrophil responses without airway obstruction or excess inflammation [54]. This contrasts with evidence in humans where excess mucus production may contribute to increased morbidity and mortality [55].

Similar to the respiratory epithelium, the small intestinal epithelium is covered in a mucus layer that segregates the commensal microbes from the epithelial cells. Goblet cells secrete mucus, mainly MUC2, in response to the presence of microbial products and Th2 cytokines, producing a thin layer of mucus that coats the microvilli [56]. Interferon-γ (IFN-γ) signaling induces Paneth cell and goblet cell secretion, resulting in more robust secretions than would occur with PAMP stimulation alone [57]. By excluding most microbes, the small intestinal mucus layer also plays an important role in maintaining immune tolerance. In contrast to the colon (which has two mucus layers), the small intestinal mucus layer is somewhat porous, which facilitates antigen sampling by various DC subsets and the generation of T regulatory cells [58]. MUC2 tolerizes DCs by inducing IL-10 secretion by epithelial cells and directly inhibits DC secretion of pro-inflammatory IL-12 [58].

## 5. Immune Response by Mucosal Epithelial Cells

### 5.1. Pathogen Detection

In addition to their role as a physical/mechanical barrier, epithelial cells are important immune players, with roles in host defense and modulation of immunity. Epithelial cells recognize PAMPs via pattern recognition receptors (PRRs), including TLRs, Nod-like receptors (NLRs), and retinoic acid-inducible gene-I-like receptors (RLRs; intracellular receptors for viral RNA, including RIG-I and MDA5). The intracellular NLR NOD1 is ubiquitously expressed in human tissues, while NOD2 is restricted to immune and epithelial cells [59]. While TLRs are ubiquitous, expression is more nuanced at mucosal epithelial surfaces to prevent immune overactivation. This is critical in the respiratory and small intestinal epithelia as both feature robust populations of commensal bacteria and other microbes, collectively known as the microbiome. TLR activation levels are managed by regulating expression levels between the apical and basolateral cell surfaces of mucosal epithelial tissues [60]. In contrast, NLRs and RLRs are inherently shielded from overactivation by their intracellular localization. Because they require tightly controlled localization, TLRs are of particular interest to mucosal immunity and will be the primary focus of this section. An overview of the expression patterns within the respiratory and small intestinal epithelium is provided in Table 2.

In general, both the respiratory epithelium and small intestinal epithelium express minimal levels of TLRs and exhibit limited transcriptional responses to ligands to maintain immune homeostasis [61,62]. In the small intestinal epithelium, TLR4 and its co-receptor MD-2 are expressed in low levels on the basolateral membrane, and are hyporesponsive to LPS, as indicated by low levels of NF-κB and CXCL8 production in response to LPS stimulation [62]. A similar pattern of expression is observed in the respiratory epithelium, which, in combination with limited MD-2 expression, limits responsiveness to LPS [63]. TLR4 has also been identified intracellularly within both tissues, where it colocalizes with internalized LPS [64]. In intestinal epithelial cells, this results in a high initial level of responsiveness to LPS, followed by a state of tolerance that is induced downstream from the receptor [64]. For both tissues, changes in the expression of TLR4 under inflammatory conditions are less clear and may depend on the activating pathogen and inflammatory signals at play [64,65,66]. More work remains to be done to fully understand TLR4 expression under homeostatic and inflammatory conditions in these tissues.

TLR2, which detects a variety of Gram-positive and Gram-negative bacterial cell wall-associated PAMPs, plays an important role in both epithelial tissues. TLR2 is constitutively expressed on the basolateral surface of the small intestinal epithelium, with minimal expression along the apical brush border, which facilitates the detection of pathogens that have invaded the small intestine [67]. The small intestinal epithelium is able to upregulate TLR2 expression in response to colonization but can become unresponsive with repeated stimulation [68,69]. In the respiratory epithelium, TLR2 is present in small amounts on the apical cell surface and is rapidly mobilized to the area in lipid raft microdomains in response to receptor activation [70]. This allows for a response to the apical detection of pathogens, rather than invasion as is required by the intestinal epithelium. This trend showing the upregulation of PRRs within the respiratory epithelium has also been observed for RLRs, RIG-I, and MDA5 [71]. In particular, IFN-γ can enhance RIG-I signaling to improve anti-viral responses [72].

Other TLRs of particular interest in the context of the respiratory and small intestinal epithelia include TLR5 and TLR7. TLR5 recognizes bacterial flagellin, which is associated with motile bacteria. Notably, TLR5 expression is restricted to the basolateral side of Paneth cells in the small intestinal crypts, providing protection from invading motile bacteria [61]. In the lower respiratory epithelium, TLR5 expression, though more widespread than in the small intestine, is also confined to the basolateral surface of the respiratory epithelium, but it can be recruited to the apical surface in response to flagellin sensing [73,74]. In contrast, TLR7 detects single-stranded RNA and is important for anti-viral responses, including the production of type 1 interferon [60]. TLR7 is ubiquitously expressed in cells of the respiratory epithelium, both intracellularly and apically, but is absent or produced at undetectable levels in the small intestine [61,73,75].

Other nucleic acid-sensing PRRs, including TLR3 (double-stranded RNA) and TLR9 (unmethylated, single-stranded DNA), have variable expression between the two tissues; they are exclusively intracellular within the small intestinal epithelium but can be intracellular or apically-expressed in the respiratory epithelium [60,76]. These differences likely represent an adaptive phenotype meant to render the respiratory epithelium significantly more sensitive to luminal pathogens than the small intestine epithelium, aligning with the expression patterns seen for other PRRs as previously described.

### 5.2. Response to Pathogen Recognition Receptor Signaling

TLR signaling ultimately results in the activation of NF-κB either through the MyD88-dependent pathway, TRIF-dependent pathway, or both [77]. The TRIF-dependent pathway also activates IRF3 that triggers IFN production for an anti-viral response. NOD1, NOD2, RIG-I, and MDA5 activation also converge on the downstream activation of NF-κB [78]. This redundancy minimizes the impact of loss of function mutations in specific PRRs. As one example, a deficiency in either TLR2 or TLR4 within the small intestine will result in increased expression of the opposing receptor to preserve function [68]. Despite this redundancy, mutations or deletions can still be associated with inflammatory conditions. Loss of function NOD2 variants are highly associated with inflammatory bowel disease (IBD) risk, which may be due to defects in barrier function resulting from strong Th1 responses [79,80,81]. In the lung, TLR4 deletion results in the upregulation of NADPH oxidase 3, resulting in heightened levels of oxidants leading to emphysema [82]. Mutations in the downstream signaling pathway molecules tend to have broader consequences on the inflammatory response and even increased susceptibility to infection due to these convergent pathways, as illustrated in various primary immunodeficiencies [83].

TLR activation by commensal organisms has an additional role in the maintenance of immune homeostasis within tissues. TLR activation contributes to the numbers of mast cells in the small intestine, and intraepithelial lymphocyte populations in the respiratory epithelium [84,85]. Of note, TLR activation on intestinal epithelial cells stimulates IL-10 synthesis, which then modulates barrier integrity (i.e., regulating E-cadherin and desmoglein-2) in an autocrine fashion [86]. This modulation is also important for local tolerance to commensal organisms [87]. IL-10 may have a similar mechanism in the respiratory tract, as changes in diet altered both the microbiome and IL-10 expression in a porcine model, but further research is needed [88]. Other PRRs also play a role in homeostasis, for example, non-canonical RIG-I activation by commensal viruses regulates intraepithelial lymphocytes in the intestine [89]. The literature describes a gut–airway axis, with crosstalk occurring between the gut microbiome and airway immunity [90]. Similarly, recent evidence also suggests that the lower respiratory microbiome can also impact the gut microbiome, with both microbiomes contributing microbiology diversity in both mucosal epithelia [91]. These types of interactions are likely to become more important in the future as technological advances allow for the detailed characterization of the microbiome in these tissues and a deeper understanding of changes that occur during diseased states.

PRR activation beyond baseline levels results in the production of chemokines to recruit additional immune cells. Neutrophils are among the earliest cells recruited to the site of infection. CXC chemokines, such as CXCL8, are potent neutrophil chemotactic molecules in both the respiratory and small intestinal epithelium [92]. The importance of neutrophil recruitment is reflected in the variety of cytokines that are involved in neutrophil recruitment, including CXCL1, CXCL2, CXCL5, and GM-CSF [93,94]. Aside from a role in neutrophil recruitment, GM-CSF is involved in myeloid and DC differentiation and survival in both epithelia [95]. The rapid secretion of cytokines in response to inflammation results in autocrine signaling that affects the release of other mediators. As an example, early secretion of TNF by respiratory epithelial cells contributes to the biphasic expression of CXCL8 expression by initially dampening and later enhancing secretion [96]. Though relatively few examples are presented here, both the respiratory and small intestinal epithelia produce multiple redundant cytokines that contribute to immune cell recruitment. Many of these cytokines are also produced by additional cell types, including local and recruited immune cells, which creates a positive feedback loop and the robust recruitment of further immune cells.

Epithelial cells are the primary source of several key cytokines within the small intestinal and lower respiratory epithelia. Thymic stromal lymphopoietin (TSLP), which is indirectly involved in T cell recruitment and is indicative of type 2 inflammation, is one example of a cytokine primarily produced by epithelial cells. TSLP, along with additional epithelial cytokines IL-33 and IL-25, are produced by bronchial epithelial cells and have a role in mediating type 2 inflammation [97]. In contrast, TSLP is induced by the microbiome in the small intestine, where it promotes homeostasis via Treg development and tolerizes DCs [98]. Short and long isoforms of the cytokine may explain the paradoxical activities of TSLP, as the long isoform is elevated in a variety of inflammatory diseases, including asthma, ulcerative colitis, and Celiac disease [98]. Research on this topic has proven challenging, as mice do not express the short isoform that is primarily homeostatic in nature. Anti-TSLP therapy appears quite effective in severe asthma and is currently in clinical trials but there remains the potential for interference with the homeostatic actions of short TSLP in the intestine or other tissues [99].

### 5.3. Anti-Microbial Peptides, Proteins, and Other Effectors

The downstream products of PRR activation include the secretion of various anti-microbial effectors, many of which are common to both the respiratory and small intestinal epithelia. Selected molecules will be discussed in this article, including anti-microbial peptides (AMPs), regenerating islet derived III (RegIII) proteins, reactive oxygen species (ROS), and antiproteases.

Epithelial cells produce small cationic peptides (i.e., alpha and beta defensins, cathelicidins, and collectins), collectively referred to as AMPs, that can disrupt negatively charged bacterial membranes resulting in microbial lysis and death. Chemokines produced by epithelial cells in response to IFN-γ have similar activity, attributed to a defensin-like domain [100]. Alpha defensins are produced by both bronchial epithelial cells and Paneth cells in response to TLR or NLR signaling, and beta-defensins are produced in a similar fashion in both tissues [101,102,103,104]. In addition to membrane-based activity, human alpha-defensin six is able to form “nanonets” that trap bacteria and prevent intestinal invasion, and may also suppress the virulence factors of some pathogens, including *C. albicans* [105,106]. In humans, both tissues express the cathelicidin LL-37, though within the small intestine expression tapers off after the neonatal period when immunocompetent Paneth cells begin producing defensins [47]. In addition to membrane disruption, LL-37 also acts as a chemoattractant for innate immune cells and can sequester LPS when in its active form [100]. Collagen-containing C-type lectins (collectins), secreted as part of a protective club cell phenotype in the respiratory epithelium and by Paneth and epithelial cells in the small intestine, have anti-membrane activity and act as soluble PRRs that agglutinate bacteria and contribute to opsonization [107,108]. Collectins and other soluble lectins have a wide expression profile across both the small intestine and respiratory tract, where they contribute to both microbial defense and maintenance of the local microbiome [109].

RegIII proteins, an example of C-type lectins, are anti-bacterial lectins that target Gram-positive bacteria by binding peptidoglycan and damaging the cell wall. RegIII proteins are produced by Paneth cells of the small intestine in response to TLR signaling, where expression correlates with microbiome diversity [110,111]. Additionally, RegIIIγ has barrier functions, physically separating the microbiota from the villi tips in the small intestinal epithelium. This limits contact between microbes and the apical surface, with enhanced contact triggering increased expression [112,113]. In the respiratory epithelium, IL-22 derived from CD4+ T cells induces the expression of RegIIIγ by goblet and club cells in the respiratory tract, where it may play a role in mediating type 2 inflammation [114,115]. Additional evidence suggests that RegIIIγ may have direct anti-viral activities, and can increase phagocyte responsiveness to respiratory viruses [116].

Reactive oxygen (ROS) and nitrogen species (NOS) have direct anti-microbial activity against membranes and DNA, in addition to various signaling capabilities. In the respiratory epithelium, dual oxidases on the apical surface of ciliated cells constitutively produce ROS, though production can also be upregulated moderately in response to IL-4 and IL-13, and significantly in response to IFN-γ [117]. The actions of ROS within the respiratory epithelium are amplified by the lactoperoxidase-catalyzed formation of highly microbicidal hypothiocyanite [118]. Dual oxidases are also expressed within the small intestinal epithelium, with expression regulated primarily by commensal segmented filamentous bacteria [119]. Commensal bacteria cause the transient generation of ROS, which modulates the ubiquitin-proteasome system to maintain inflammatory tolerance via regulating levels of NF-κB [120,121]. Additionally, NOD2 signaling in response to pathogen detection results in epithelial cell ROS production at increased levels for anti-microbial defense [122]. Despite the important roles of ROS in mucosal immunity, excessive levels of oxidative stress are harmful to the host and underly certain pathologies [123].

Epithelial cells produce antiproteases, specifically the secretory leukocyte protease inhibitor (SLPI) and elastin-specific inhibitor (elafin), in response to pro-inflammatory signals. Both SLPI and elafin exert additional anti-microbial effects on various Gram-positive and Gram-negative bacteria, and anti-inflammatory effects on macrophages and endothelial cells [100,124]. The protease/antiprotease balance must be tightly regulated, as an imbalance can increase susceptibility to infection [125]. In cystic fibrosis, the antiprotease system is overwhelmed by neutrophil proteases, which results in inflammation and impaired immune responses [126]. Elafin and SLPI are also expressed in the small intestine but few studies have addressed the specific roles of antiproteases here [127,128]. Dysregulated levels of antiproteases may contribute to small intestinal pathology; gluten-intolerant patients have reduced elafin production, leading to enhanced tissue damage [128]. This illustrates the importance of protease/antiprotease balance within the small intestine, but more research is needed to clarify the specifics of this in the case of disease.

Epithelial cells can produce other effector molecules that indirectly affect pathogens. Iron sequestration by molecules like lactoferrin and lipocalin-2 negatively impacts bacterial growth and prevents excessive inflammation via iron-catalyzed ROS generation. Lactoferrin is expressed by epithelial cells within the respiratory tract and the small intestine and by neutrophils. It has been proposed as a therapy for the respiratory virus SARS-CoV-2 due to anti-inflammatory and anti-microbial properties, which may reduce the development of secondary bacterial infections [129]. Though typically bacteriostatic, lactoferrin is bactericidal when acting in combination with lysozyme, which is abundant in both the respiratory tract and small intestine [130,131]. Lipocalin-2 has a similar role with respect to iron sequestration, and is expressed by goblet cells in the lower respiratory tract and by a subset of Paneth cells in the small intestine [132].

Though most anti-microbial effector molecules are shared between the two epithelia, there are some tissue-specific molecules. The respiratory epithelium produces proteins from the palate, lung, and nasal epithelium clone (PLUNC) family, which are related to both LPS-binding protein and bactericidal/permeability increasing protein. PLUNC proteins are multi-functional, with surfactant properties and some specific anti-bacterial and anti-biofilm effects [133,134]. Expression of these proteins appears limited to the respiratory epithelium, with strongest expression within the trachea [135]. As an example, bactericidal/permeability-increasing protein fold-containing family member A1 (BPIFA1) is abundant in the respiratory tract and has a variety of anti-microbial, surfactant, and immunomodulatory properties [135]. PLUNC proteins have not been identified within the small intestine, though there are low levels of expression within the stomach [136].

## 6. Impact of Innate Immune Activation on the Epithelium

Innate immune activation can affect the epithelium in a variety of ways beyond the previously described anti-microbial responses. For example, Th2 signaling with IL-4 and IL-13 results in the expansion of specific cell populations within both the respiratory and small intestinal epithelia [137,138]. This section of the article will focus primarily on the response of the epithelium to cytokine expression in terms of changes in proliferation and barrier permeability. The effects of these cytokines on the lower respiratory and small intestinal epithelia are outlined in Table 3.

Secreted cytokines can significantly affect epithelial cell proliferation in both the respiratory tract and small intestine. Though tissue renewal and repair are critical functions, alterations to these processes can result in increased proliferation (hyperplasia) and pathology. Though best described in the colon, benign but abnormal proliferation also occurs within the small intestine in IBD [139,140]. Hyperplasia can also occur within the respiratory tract. In chronic obstructive pulmonary disease patients, basal cell hyperplasia and squamous cell metaplasia are associated with reduced epithelial permeability and susceptibility to infection [141]. Reduced proliferation, leading to hypoplasia, is less common but can also have significant negative effects. Impaired respiratory epithelial proliferation is associated with bronchiolitis obliterans, which impairs pulmonary function and may lead to respiratory failure [142]. The differences in the impact of these proliferative changes between the epithelia are likely related to inherent structural and functional differences. The length and functions of the small intestine may allow it to tolerate substantially more damage than the respiratory tract, where an obstruction can quickly become fatal. The effects of selected cytokines on the two epithelia, including IFNs, transforming growth factor-β (TGF-β), IL-4, IL-13, and IL-6, will be reviewed in the coming sections and are summarized in Table 2.

IFNs, produced by both innate immune cells and epithelial cells in response to viral infection, have diverse actions within the respiratory and small intestinal epithelia, with type I and type III IFNs playing a key role in anti-viral immunity. Type I IFNs (i.e., IFN-α and IFN-β) have anti-proliferative and pro-apoptotic effects within the small intestine, suggesting involvement in the maintenance of barrier homeostasis [143,144]. Type I IFNs also impair epithelial proliferation within the lungs during recovery from viral infection [145]. Type III IFNs (i.e., IFN-λ) have significant functional overlap with type I IFNs, though they are less inflammatory in vivo than either IFN-α or IFN-β [146]. Despite the reduced inflammatory effects associated with IFN-λ, chronic exposure can result in negative effects, as IFN-λ reduces epithelial proliferation and hinders epithelial repair in the lower respiratory tract [145]. Data related to the effects of IFN-λ signaling within the small intestine are limited, but in the colon, both anti-proliferative effects and enhanced mucosal healing have been observed, suggesting that exact effects may be situational [147,148].

Type II IFN (IFN-γ) is also known to affect cell proliferation within the small intestinal and lower respiratory epithelia, and these effects are dependent on the duration of the exposure. A brief exposure induces proliferation while an extended exposure inhibits proliferation by depletion of the Wingless-Int co-receptor required for beta-catenin signaling pathways [149]. Paneth cells expansion, mediated by altered β-catenin signaling, has also been identified in response to *Salmonella enterica* and *Trichinella spiralis* infection [150,151]. This adaptive response, most likely mediated by IFN-γ, facilitates both inflammatory responses and delivery of anti-microbial effector molecules. Continuous IFN-γ exposure results in the progressive loss of Paneth cells, leading to rapidly cycling Ki-67+ cells within the intestinal crypts and hyperplasia [57]. This has been replicated experimentally; IFN-γ administration leads to enteropathy and crypt hyperplasia [152]. However, these effects appear to be dependent on the situation, as IFN-γ and IL-4 both disrupted respiratory epithelial barrier function in vitro via the downregulation of the expression of tight junction proteins [153]. There is additional evidence to suggest that IFN-γ causes barrier damage by inhibiting proliferation and increasing apoptosis in the small airway [154]. These differences may be related to the duration of exposure, as seen in the small intestine, or may be dose-dependent.

TGF-β, produced by many immune cells, affects proliferation by regulating the expression of survivin. TGF-β strongly suppresses survivin, which is an inhibitor of apoptosis, to maintain tissue homeostasis and prevent tumor formation. This pathway is disrupted in some tumors, leading to high levels of survivin and TGF-β [155]. TGF-β also has pro-apoptotic effects, in addition to being a negative regulator of surviving [156]. Knockdown of TGF-β in an inflammatory environment is sufficient to cause invasive colon cancer and small intestinal polyps in mice [157]. Additional roles in the maintenance of small intestinal barrier integrity have also been described [158]. The actions of TGF-β appear to be relatively conserved between the two epithelia, with TGF-β signaling also playing an important role in epithelial homeostasis of the small airway [159]. Dysregulated signaling has been linked to increased permeability during acute lung injury and may play a role in the development of edema and fibrosis [160]. Experimentally, receptor knockdown provided mice with protection against fibrosis in a model of lung injury [161].

Th2 cytokines, IL-4 and IL-13, are primarily associated with proliferative effects in both tissues. This typically represents an adaptive response to infection, with increased goblet cell proliferation upregulating the expression of mucins and several AMPs [138]. This can become maladaptive, with excess mucus impairing mucociliary clearance [162]. Goblet cell hyperplasia frequently occurs with chronic allergen exposure where Th2 cytokines persist for prolonged periods [163]. Similarly to its actions in the respiratory tract, IL-4 enhances intestinal epithelial cell proliferation in a dose-dependent manner and supports the expansion of specific cell populations [164]. Small intestinal tuft cell and goblet cell differentiation are induced by IL-4 and IL-13 produced during helminth infections, which helps maintain mucus levels required for worm expulsion [34]. IL-33, another Th2 cytokine, is associated with similar changes in cell populations in the small intestine [165].

IL-6 mediates a variety of effects within the small intestine and the lower respiratory tract via distinct signaling pathways. The classical (cis) pathway is involved in the antibacterial and anti-inflammatory effects of IL-6, while trans signaling, in which IL-6 binds to the soluble IL-6 receptor associated with gp160 in cell membranes, is linked to pro-inflammatory activities [166]. These pathways are illustrated in Figure 2. In the intestine, Paneth cells promote intestinal homeostasis via autocrine cis IL-6 signaling, resulting in proliferation and repair [167]. Trans signaling, which allows IL-6 to act on cells that do not express the receptor, leads to inflammation and hyperplasia of the small intestinal crypts [168]. Trans signaling is also strongly associated with cancer; excess epidermal growth factor increases receptor activation, resulting in IL-6 production and further aberrant proliferation [169]. In the lung, trans signaling also contributes to fibroblast proliferation and increased extracellular matrix production [170]. Based on these examples, a specific blockade of trans IL-6R signaling has been proposed as a potential therapy for intestinal hyperplasia, lung cancer, and pulmonary fibrosis [170,171].

Many additional cytokines are able to act on the small intestinal and respiratory epithelium. Andrews et al. (2018) provide an in-depth review of the effects of different cytokines on the intestinal epithelium, while Branchett et al. (2019) review regulatory cytokines in the respiratory tract [172,173]. It is worth noting that most data available address the impact of these cytokines on the colon epithelium, and findings may not be transferable to the small intestine due to differences in cell populations (i.e., Paneth cells), microbiome, and general function.

## 7. Lessons from the Respiratory and Small Intestinal Epithelia

The lower respiratory tract and small intestine, despite differences in structure and cell function, display remarkable similarities in terms of host defense. The epithelia had comparable responses to pathogen detection and secreted cytokine mediators. For the most part, both tissues produced comparable anti-microbial effector molecules. One distinction is PRR localization and the response to cytokines produced by infiltrating cells. Both the similarities and differences observed between the epithelia present opportunities for potential therapies that could either be useful for mucosal immunity in a variety of tissues, or epithelium-specific therapies.

The overlying mucus layer found in both tissues both acts as a physical barrier against pathogens and plays a direct role in the immune function. In both epithelia, mucins can be upregulated in response to PAMPs. Different mucin molecules reportedly have specific immune functions, which may be tissue-specific. MUC5AC had a protective effect in the respiratory epithelium, but in humans, excess mucus production is thought to be pathological. This likely limits the utility of mucin-based therapies in the respiratory tract. This strategy may still be viable in the small intestine. Specifically, MUC2 plays an important role in immune tolerance. MUC2 deficiencies have been associated with low levels of chronic inflammation, altered tissue homeostasis, and tumor development in the small intestine [174]. As administration of MUC2 can correct these deficiencies, this may represent a potential therapeutic avenue for reducing inflammation and restoring immune tolerance within the gut [58].

Both tissues have evolved to tolerate the presence of commensal organisms while also preventing pathogens from colonizing the tissues. It should be noted that the presence of the same functional outcome or phenotype in the two epithelia does not necessarily indicate that the same signaling pathway is involved. The GI microbiome is robust and reasonably well-characterized, as are various mechanisms of tolerance, such as DC sampling and TLR hyperresponsiveness. In the respiratory tract, knowledge of the microbiome is rapidly developing due to advances in single-cell sequencing techniques. Mechanisms of tolerance may be shared, including regulation of antigen-specific tolerance by DCs, but more research is needed to fully understand these mechanisms in the respiratory tract [175]. Despite differences in the composition of the microbiome and potentially the mechanisms of immune tolerance in the respiratory and small intestinal epithelia, modulation of these pathways may represent a future therapeutic option. This could occur through the direct modulation of microbiome composition or indirectly via administration of key molecules (of host or microbial origin) involved in mediating tolerance. RegIIIγ is expressed in both tissues but it serves as a barrier in the small intestine that physically separates the microbiota from the epithelium. In the intestine, diet can alter ileal expression of RegIII proteins, which is at least partially mediated by changes in microbiome composition and the number of Bifidobacteria present [176]. Fecal microbial transplants or administration of Bifidobacterium probiotics may be able to restore immune homeostasis in the small intestine. Similar barrier functions have not been described in the respiratory tract but the administration of RegIIIγ, indirectly via secretion from adipose-derived stem cells, was protective against *S. aureus*-induced lung injury [177]. Key bacteria responsible for RegIIIγ production within the lung have not yet been described but could represent a therapeutic avenue for regulation of immune function in the lung. Direct administration of RegIIIγ could be an additional strategy, though this would not address any underlying issues with microbiome composition.

One key difference between the respiratory and small intestinal epithelium is the expression of PRRs. The respiratory tract appears to have more apically expressed TLRs than the small intestine. This is likely an adaptive response, allowing for a more rapid response to pathogens than can occur in the small intestine. This distribution may reflect the fact that PRR detection of PAMPs in the respiratory tract means that pathogens have escaped initial defenses, including mucociliary clearance. A rapid response is critical, due to the importance of the respiratory tract to immediate survival. Additionally, the respiratory epithelium appears to have an increased ability to recruit TLRs to the apical surface in response to inflammatory signals, as is observed with TLR2 [70]. In the intestine, TLRs are largely compartmentalized to the basolateral membrane, with the exception of TLR2 expressed apically within follicle-associated epithelia. As TLR2 is expressed on the apical surfaces of both tissues, TLR2 blockades may be useful for reducing hyperinflammation caused by Gram-positive pathogens when combined with appropriate antibiotics to adequately manage infection. TLR2 blockade, mediated by the p53 protein, may also reduce inflammatory responses to respiratory syncytial virus [178]. These therapies must be employed with caution; they may negatively affect immune homeostasis, as commensal organisms within the small intestine are known to maintain tolerance via TLR2 signaling [179]. Additionally, TLR2 agonism can also have an immune priming effect within the airway, enhancing responsiveness to viral infections and reducing local inflammation [180]. These findings suggest that the role of TLR2 is likely context-specific. Modulation of these responses therapeutically requires significant investment into the disease-specific TLR function.

Disruptions in epithelial cell polarity, including trafficking of proteins to inappropriate membranes, can have various negative health consequences that are likely related to aberrant TLR signaling and increased barrier permeability. Changes in epithelial permeability are well-described in the colon of IBD patients and have also been noted in pulmonary fibrosis [67,181]. In intestinal epithelial cells, defects in apical protein localization lead to malnutrition, while those affecting basolateral proteins result in architectural issues and cancer [182]. Injuries to epithelial cells that compromise polarity can also facilitate microbial colonization and invasion, as seen with the opportunistic pathogen *Pseudomonas aeruginosa* [183]. Therapies aimed at restoring barrier integrity, such as promoting the expression of proteins associated with cell-to-cell adherence or restoring lost polarity, may be useful in the management of *P*. *aeruginosa* infections. As an example, an integrin-activating monoclonal antibody re-established contact between the basolateral surface and the extracellular matrix, restoring epithelial polarity in invasive colon cancer.

The lower respiratory and small intestinal epithelia have very similar patterns of cytokine expression, particularly in terms of chemokines expressed in response to pro-inflammatory signals. TSLP provides an example of a cytokine produced by both epithelia that results in different effects in each tissue. As reviewed by Ebina-Shibuya and Leonard, TSLP is pleotropic by nature, and can have beneficial effects (i.e., host–pathogen defense) and negative effects (i.e., allergic disease), which are likely related to the specific isoform expressed. In the small intestine, TSLP overexpression is associated with atopy, but TSLP knockdown compromises T cell-mediated immune tolerance to commensal microbes [184]. In contrast to this, TSLP expression in the respiratory tract can impair pulmonary Treg function, ultimately enhancing airway inflammation [185]. Though more work needs to be done to understand the expression patterns of the short and long isoforms of TSLP in disease states and their specific mechanisms of action, the timed targeting of specific isoforms may be useful in a variety of inflammatory conditions.

IFN-λ serves as a second example of a cytokine with potential therapeutic effects in the two mucosal tissues. As intestinal epithelial cells appear to respond preferentially to type III IFNs, IFN-λ therapies are especially interesting in this tissue [186]. Evidence suggests that IFN-λ plays an important role in defense against enteric viruses, and may be less inflammatory than other interferons, but many questions remain about the exact mechanisms at play [187]. The duration of treatment may be critical here, as chronic IFN-λ exposure decreases epithelial repair. This is of more concern in the respiratory tract, which has a much slower rate of epithelial regeneration. IFN-λ may still be of use for the control of viral infections in the respiratory tract, as a single injection of “peginterferon-λ” administered to a small cohort of patients with SARS-CoV-2 resulted in increased viral clearance compared to placebo [188]. Recently, IFN-λ has been shown to reduce viral load and dissemination in a murine model of human metapneumovirus, while also limiting inflammatory responses and signs of clinical disease [189]. This suggests that IFN-λ may serve as a useful therapeutic in respiratory infections, particularly for viral control, but may require short durations of use to manage negative effects on epithelial repair.

## 8. Conclusions

In summary, the evidence presented here with respect to the lower respiratory and small intestinal epithelia suggests that many targeted therapies may be applicable for both tissues. The epithelia have evolved similar mechanisms to serve the same end goal with respect to microbial tolerance and host defense. The respiratory and small intestinal epithelia both need to tolerate commensal organisms while mounting an appropriate response to invading pathogens. This delicate balance is maintained by the microbiota associated with each epithelium and the compartmentalization of PRRs. Advanced sequencing techniques have been instrumental in characterizing both the cell populations within the epithelia, and the associated microbiota. As knowledge about the respiratory microbiome and its interactions with the immune system advances, new avenues for therapeutics are likely to emerge. This may be related to the reconstitution of a microbiome altered by disease states or the administration of prebiotics or specific mediators related to immune tolerance, such as RegIIIγ. Maintenance of barrier integrity is also critical to both the respiratory and small intestinal epithelia, to ensure epithelial cell polarization and trafficking of PRRs. Despite these similarities, there are still some opportunities for tissue-specific therapies, such as TSLP and IFN-λ. More opportunities for tissue-specific therapies are likely to emerge in the future as further advances are made.

## Figures and Tables

**Figure 1 biomedicines-13-01052-f001:**
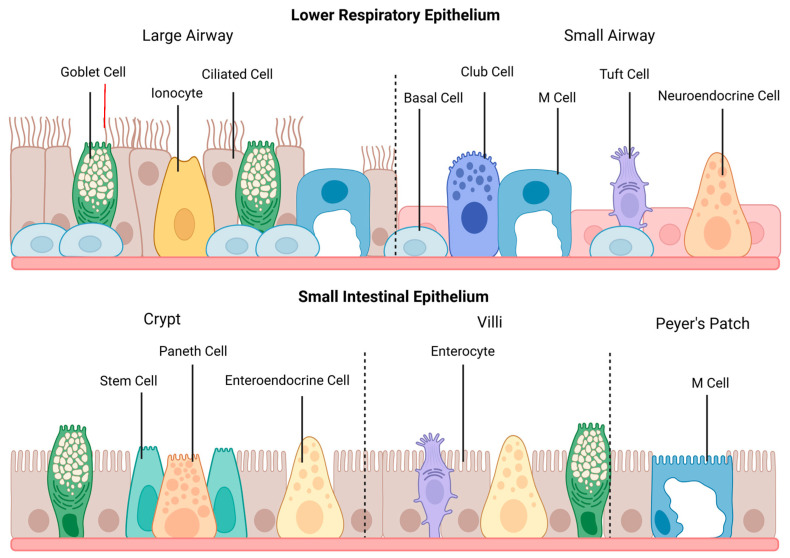
The general structure and cell populations of the lower respiratory and small intestinal epithelia. The lower respiratory epithelium is subdivided into the large and small airways. Ciliated cells are the primary cell type within the large airway, which also contains goblet cells, pulmonary ionocytes, and M cells. The epithelium transitions to simple cuboidal cells within the small airway, and goblet and basal cells become less numerous. This section of the lower respiratory tract also contains club cells, tuft cells, and neuroendocrine cells. M cells may occur during pathological conditions featuring lymphoid infiltration. The small intestinal epithelium shows cell populations from the crypts, villi, and Peyer’s patches. Enterocytes are the primary cell type within the small intestine, and goblet cells and enteroendocrine cells are featured throughout the epithelium. Paneth cells, surrounded by stem cells, are located in the intestinal crypts, and tuft cells are found on the villi. M cells are primarily found above Peyer’s patches. This figure was created with Biorender.com.

**Figure 2 biomedicines-13-01052-f002:**
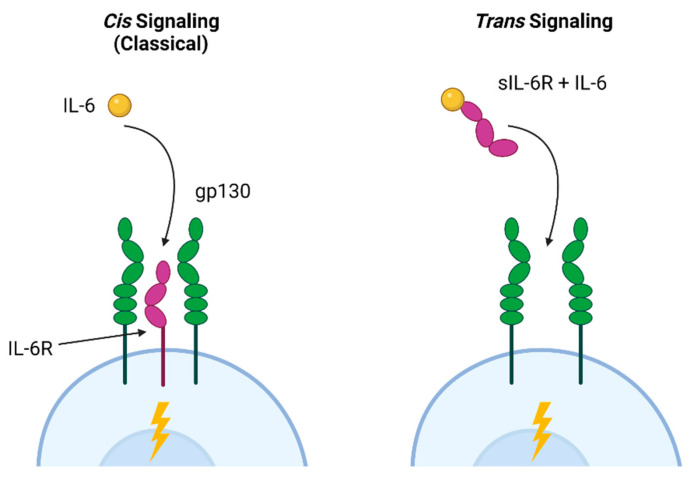
Cis and trans IL-6 signaling pathways. Classical cis IL-6 signaling, associated with homeostasis and anti-inflammatory activities, occurs when the IL-6 ligand binds to IL-6R and co-receptors (gp130) on a cell. This allows cells expressing IL-6R to signal in response to ligand binding. Trans signaling occurs when IL-6 binds to the soluble (sIL-6R) receptor that was shed into the extracellular environment in response to inflammatory signals. This complex can then bind to any gp130-expressing cell, leading to IL-6 responsiveness in cells that are not expressing IL-6R. Figure created with Biorender.com.

**Table 1 biomedicines-13-01052-t001:** An overview of major cell types found within the lower respiratory and small intestinal epithelia, classified according to their primary function.

	Epithelium	
Cell Function	Respiratory	Small Intestinal
Primary cell type	Ciliated	Enterocytes
Regeneration	Basal, Club Deuterosomal, KRT13+ (Hillock)	Intestinal stem
Chemosensory	Tuft	Tuft (1/2)
Mucus secretion	Goblet	Goblet
Secretory cells	Ionocytes, Club	Paneth
Antigen sampling	M	M
Endocrine cells	Pulmonary neuroendocrine	Enteroendocrine

**Table 2 biomedicines-13-01052-t002:** PRR expression within the respiratory and small intestinal epithelia.

Receptor		PAMP	Epithelium					
			Respiratory			Small Intestinal		
			Apical	Baso	Cyto	Apical	Baso	Cyto
TLRs	TLR4	LPS		+	++		+	++
	TLR2	Lipoproteins, LTA	++ *			+	++ *	
	TLR5	Flagellin	+ *	++			++	
	TLR7	ssRNA		+	+		-	
	TLR3	dsRNA	+		+			+
	TLR9	ssDNA	+		+			+
NLRs	NOD1	γ-D-*meso*-DAP			+			+
	NOD2	Muramyl Dipeptide			+			+
RLRs	RIG-I	Viral RNA			+ *			+
	MDA5	Viral RNA			+ *			+

Note: Levels of expression are indicated qualitatively, with ++ representing higher levels relative to +. The presence of * denotes changes in expression in response to activation. Baso—basolateral; Cyto—cytoplasmic.

**Table 3 biomedicines-13-01052-t003:** Effects of cytokine mediators on respiratory and small intestinal epithelial function (↑ Up is increased and ↓ down is decreased).

	Epithelium	
Mediator	Lower Respiratory	Small Intestinal
Type I IFNs	↓ proliferation	↓ proliferation
Type II IFNs (IFN-γ)	↑ DUOX and RIG-I ↓ proliferation ↑ apoptosis ↓ barrier function	↑ proliferation (acute exposure) ↓ proliferation and loss of Paneth cells (chronic exposure) ↑ Paneth/goblet cell secretion
Type III IFNs (IFN-λ)	↓ proliferation ↓ epithelial repair (chronic exposure)	↓ proliferation ↑ mucosal healing
TGF-β	↓ proliferation ↑ apoptosis Maintenance of barrier permeability	↓ proliferation ↑ apoptosis Maintenance of barrier permeability
IL-4/IL-13	↑ mucin/AMP secretion ↑ goblet cell proliferation (chronic exposure) ↓ decreased lung function (long-term)	↑ mucin/AMP secretion ↑ tuft/goblet cell differentiation
IL-6	↑ proliferation (trans signaling) ↑ fibrosis (trans signaling)	Homeostasis and wound healing (classical signaling) ↑ proliferation (trans signaling)
Type I IFNs	↓ proliferation	↓ proliferation

## Data Availability

No new data were created or analyzed in this study. Data sharing is not applicable to this article.

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
