# Peer review of "Mucosal Immunity: Lessons from the Lower Respiratory and Small Intestinal Epithelia"

_biomedicines, 2025, doi:10.3390/biomedicines13051052_

Round 1

Reviewer 1 Report

Comments and Suggestions for Authors

The authors provide an extensive description of morphology, physiology, defence systems, and immune responses in two mucosae areas and consequently in two different types of epithelium, airway epithelium and small intestine epithelium. 
The manuscript covers these structures' physiology and immunological role, but the morphological description needs some clarification. As a result, I have the following suggestion for the authors:
-    Kindly revise the Figure 1. The goblet cell is the green cell. Please also add the M cell in this figure;
-    The intestinal crypt epithelium also includes goblet cells in the small intestine, added to endocrine cells (only three types of cells are described in Figure 1).
-    Goblet cells are found only in the first part of the small airway epithelium. After that, they are replaced with club cells. Please explain this specific structure in the main text and Figure 1
-    Kindly expand the data regarding deuterosomal cell differentiation (please see   Ruysseveldt et al., 2021 https://doi.org/10.3389/falgy.2021.787128)
-    Please revise the text in line 413
-    Kindly add references in the Introduction and General Structure and Function Section (The first reference starts in line 81).

Author Response

We thank the reviewer for their helpful comments on our manuscript. Please find our point-by-point responses described below and highlighted within the text.

Comment: Kindly revise the Figure 1. The goblet cell is the green cell. Please also add the M cell in this figure

Response: Thank you for the note. We have revised this figure to properly reflect the goblet cell and added M cells to both epithelia.

Comment: The intestinal crypt epithelium also includes goblet cells in the small intestine, added to endocrine cells (only three types of cells are described in Figure 1).

Response: We have also revised the figure to reflect this comment. We have also updated the figure legend to more clearly reflect the cell types shown. 

Comment: Goblet cells are found only in the first part of the small airway epithelium. After that, they are replaced with club cells. Please explain this specific structure in the main text and Figure 1.

Response: Please accept our apologies for this oversight. We have reviewed the main text to ensure club cells are clearly explained. We have also added an additional reference about club cell biology.

Comment: Kindly expand the data regarding deuterosomal cell differentiation (please see   Ruysseveldt et al., 2021 https://doi.org/10.3389/falgy.2021.787128) 

Response: We thank the reviewer for providing this useful review. We have added a reference that describes how cell cycle variants result in cell differentiation rather than division.

Comment: Please revise the text in line 413

Response: Thank you for the note. We have adjusted this heading as highlighted in the revised manuscript. 

Comment: Kindly add references in the Introduction and General Structure and Function Section (The first reference starts in line 81).

Response: Thank you for the comment. We have added some supporting literature reviews to these sections. 

Reviewer 2 Report

Comments and Suggestions for Authors

This review provides a comprehensive comparison of mucosal immunity in the lower respiratory and small intestinal epithelia, emphasizing their structural, cellular, and functional similarities and differences. The authors highlight key defense mechanisms, immune responses, and therapeutic opportunities arising from these comparisons. The manuscript is well-organized, integrates recent advancements (e.g., single-cell sequencing, organoid models), and offers valuable insights into mucosal immunity. However, some sections could benefit from deeper analysis or condensation to enhance clarity and impact.

Major comments:

While the review synthesizes existing knowledge effectively, the novel contribution lies in juxtaposing two distinct mucosal tissues to identify shared and unique therapeutic targets. This approach is commendable but risks superficiality. For example, the discussion of tuft cells and RegIII proteins could be expanded to clarify their tissue-specific roles.

The proposed therapies (e.g., TLR2 blockade, TSLP isoform targeting) are intriguing but require more nuanced discussion of challenges (e.g., balancing immune homeostasis vs. hyperinflammation).

Figure 1 (cell populations) and Table 1 (cell type comparisons) are informative but lack sufficient detail in legends (e.g., functional annotations for Hillock cells).

The reference list is extensive and mostly up-to-date, but some sections (e.g., microbiome interactions) rely on older citations (e.g., 2016–2018). Incorporating 2023–2024 studies (e.g., recent advances in respiratory microbiome characterization) would enhance relevance.

The manuscript is quite lengthy, and some concepts (e.g., PRR expression, epithelial repair mechanisms) are repeated across sections. A more concise organization—perhaps integrating immune sensing and effector functions into a single section—might improve readability.

Minor comments:

Some key statements, especially in the conclusions about therapeutic applications, would benefit from more recent citations (e.g., for IFN-λ therapy trials).

Be consistent with terms like “airway epithelium,” “respiratory epithelium,” and “bronchial epithelium.”

Clarify “apical” vs. “basolateral” expression in the context of polarized epithelia early on, especially for readers from adjacent fields.

Several minor typos and formatting inconsistencies (e.g., “hyporesponsiveness” vs. “hypo-responsiveness”; spacing in tables) should be corrected before publication.

Author Response

Comment: The proposed therapies (e.g., TLR2 blockade, TSLP isoform targeting) are intriguing but require more nuanced discussion of challenges (e.g., balancing immune homeostasis vs. hyperinflammation).

Response. We strongly agree with this comment, and have revised the text as highlighted to include some updated discussion.

Comment: Figure 1 (cell populations) and Table 1 (cell type comparisons) are informative but lack sufficient detail in legends (e.g., functional annotations for Hillock cells).

Response: Thank you for this note. We have added some additional detail to the figure and have updated the table to clearly refer to Hillock cells.

Comment: The reference list is extensive and mostly up-to-date, but some sections (e.g., microbiome interactions) rely on older citations (e.g., 2016–2018). Incorporating 2023–2024 studies (e.g., recent advances in respiratory microbiome characterization) would enhance relevance.

Response: Thank you for your comment. We have conduced an additional literature search and have added an updated review on the respiratory microbiome, highlighting its interactions with the gut microbiome.

Comment: The manuscript is quite lengthy, and some concepts (e.g., PRR expression, epithelial repair mechanisms) are repeated across sections. A more concise organization—perhaps integrating immune sensing and effector functions into a single section—might improve readability.

Response: Thank you for the suggestion. We have reviewed the manuscript to look for instances of repeated information to help condense the information. Some information has been moved from Section 6 to earlier sections to keep individual sections focused. We have also rephrased some sentences to improve the overall word economy of the manuscript. 

Comment: Some key statements, especially in the conclusions about therapeutic applications, would benefit from more recent citations (e.g., for IFN-λ therapy trials).

Response: We agree with this, and have added some more recent references here and updated the discussion to highlight potential challenges with epithelial repair. 

Comment: Be consistent with terms like “airway epithelium,” “respiratory epithelium,” and “bronchial epithelium.”

Response: Thank you. We have revised instances of "airway epithelium" to say "respiratory epithelium". We have retained some instances of "bronchial epithelial cells" as they refer to specific primary literature cited. 

Comment: Clarify “apical” vs. “basolateral” expression in the context of polarized epithelia early on, especially for readers from adjacent fields.

Response: Thank you for the comment. We agree that this is an important distinction, and have moved this paragraph to earlier in the introductory section to clarify this concept sooner. 

Comment: Several minor typos and formatting inconsistencies (e.g., “hyporesponsiveness” vs. “hypo-responsiveness”; spacing in tables) should be corrected before publication.

Response: We appreciate this note, and have gone over the paper to identify any further issues. 

Round 2

Reviewer 1 Report

Comments and Suggestions for Authors

The quality of the manuscript has improved with more details and clarifications. However, I have only one suggestion for the authors. Kindly clarify the structure of the small airway epithelium in Figure 1 (including legend and main text). According to current research, M cells are not found in this epithelium under normal conditions. However, M-cells have been described in small airway epithelium associated with lymphoid infiltration in pathological conditions in murine models (as illustrated in lines 200-203). 

Author Response

We thank the reviewer for their constructive comments.

Comment: Kindly clarify the structure of the small airway epithelium in Figure 1 (including legend and main text). According to current research, M cells are not found in this epithelium under normal conditions. However, M-cells have been described in small airway epithelium associated with lymphoid infiltration in pathological conditions in murine models (as illustrated in lines 200-203). 

Response: Thank you for pointing out this oversight. We have revised the figure legend and main text to be in line with this.